# Optimal Sensor Placement for Enhanced Efficiency in Structural Health Monitoring of Medium-Rise Buildings

**DOI:** 10.3390/s24175687

**Published:** 2024-08-31

**Authors:** Salman Saeed, Sikandar H. Sajid, Luc Chouinard

**Affiliations:** 1National Institute of Urban Infrastructure Planning, University of Engineering & Technology, Peshawar 25000, Pakistan; salmansaeed@uetpeshawar.edu.pk; 2Civil Engineering, McGill University, Montreal, QC H3A 0C3, Canada; sikandar.sajid@mail.mcgill.ca; 3Civil Engineering, University of Engineering & Technology, Peshawar 25120, Pakistan

**Keywords:** modal analysis, large-scale modal testing, torsional mode, testing of medium-rise building, ambient vibration testing

## Abstract

Output-only modal analysis using ambient vibration testing is ubiquitous for the monitoring of structural systems, especially for civil engineering structures such as buildings and bridges. Nonetheless, the instrumented nodes for large-scale structural systems need to cover a significant portion of the spatial volume of the test structure to obtain accurate global modal information. This requires considerable time and resources, which can be challenging in large-scale projects, such as the seismic vulnerability assessment over a large number of facilities. In many instances, a simple center-line (stairwell case) topology is generally used due to time, logistical, and economic constraints. The latter, though a fast technique, cannot provide complete modal information, especially for torsional modes. In this research, corner-line instrumented nodes layouts using only a reference and a roving sensor are proposed, which overcome this issue and can provide maximum modal information similar to that from 3D topologies for medium-rise buildings. Parametric studies are performed to identify the most appropriate locations for sensor placement at each floor of a medium-rise building. The results indicate that corner locations at each floor are optimal. The proposed procedure is validated through field experiments on two medium-rise buildings.

## 1. Introduction

Structural health monitoring (SHM) is ubiquitously used to evaluate the global dynamic characteristics of a system using data-driven [1,2], statistical [3,4] or physics-based approaches [5]. The basic premise of SHM is to evaluate the corresponding signatures from the vibration data for comparison with baseline information and calibrate a numerical model to quantify the health of a structure [6]. For greater efficiency, output-only modal testing using ambient vibrations is a popular technique to obtain dynamic characteristics of structural systems for SHM given its efficiency and minimal interruptions of normal activities. There is a vast literature on SHM for condition monitoring and quality assurance for the reinforced concrete structures, particularly, bridges [7,8,9,10,11,12]. In all these applications, sensing nodes are selected for vibration measurement and onward processing using either data-driven or physics-based methods. The desired outcome in each approach is to obtain the maximum information on the health of the structure with greater measurement efficiency.

Practical questions when performing such measurements are the proper layout of sensors, the range of natural frequencies to be estimated, and time required for obtaining the recordings. The synchronization of measurements can also be an issue for medium and high-rise buildings, which can significantly influence the accuracy of dynamic property estimates [13,14]; however, several algorithms had been proposed to address the latter issue [15,16,17]. The layout of measurements can be a major challenge in complex structural systems. For instance, in most buildings, only public areas, such as staircases, are accessible for positioning sensing nodes. However, staircases are typically centrally located, and information obtained along these do not provide information on torsional modes [18], which are required by many national building codes (e.g., National Building Code of Canada (NBCC) 2020) when performing dynamic analyses [19].

To capture torsional modes as well as the translational modes in output-only modal testing, it is recommended to locate measurement nodes as far from the center of rigidity as possible. This is also beneficial, since torsional motions are usually of low amplitude, which can be significantly influenced by the small signal-to-noise ratios of the ambient vibration measurements in output-only modal testing. Sensing nodes to capture such dynamic behavior can be complex and potentially require longer periods for the measurements compared to the staircase measurement nodes. If the measurements are performed using an array of sensors in a network, optimal topologies have been proposed to obtain modal information with high accuracy [20,21,22,23].

In most studies on output-only modal testing, two sensing nodes are deployed on each floor to develop frame-line models [24,25,26]. In other cases, when the floor area is fully accessible, and accurate and detailed assessments of the behavior of building are required, sensors are placed on the outer periphery of the building on several floors [16]. The resulting modal model is a 3D frame that completely describes the motions of any location on the actual building.

In this research, the procedure is designed to be economical, and it is demonstrated with the use of only two sensors by selecting optimal instrument placement in a multi-step sub-structuring approach to generate dynamic characteristics of the structure with the desired level of detail. A 3D numerical model is used to simulate ambient vibration noise conditions to evaluate instrument deployment schemes and to estimate and compare the mode shapes and natural frequencies to those from the numerical model. The optimal placement scheme is identified by considering the number of sensors used, the time required for performing the measurements, and the accuracy of the mode shapes relative to the full model. The results show that the 3D behavior of a building can be obtained by assuming rigid floors and by using placement scheme for a line model where the nodes are located at the outer limit of the building. This type of topology is defined as a corner-line model. The mode shapes obtained from the proposed measurement scheme are nearly identical to those obtained from a fully measured model which includes the torsional modes that are not normally detected with the commonly used center-line model (or stairwell model). The corner-line model has been implemented in some experimental studies for symmetric structures but was not compared to estimates obtained with the frame-model [27]. The proposed method is first analyzed using numerical simulations and parametric studies, and it is subsequently applied and validated with two existing medium-rise buildings.

## 2. Numerical Models

Two numerical models of buildings were created for the analysis: (1) a symmetric model and (2) an asymmetric model. Both models have five stories with a fixed base. Floors are assumed to be rigid, and the central node of each floor is defined as a master node. The mass of the floor is assumed to be uniformly distributed and lumped at the nodes, while the beams and columns are assumed to be weightless. Out-of-plane rotations and translations of the floors were constrained.

The symmetric building is rectangular is shape, while the asymmetric building is L-shaped. Both have eight slave nodes at the periphery of each floor. The conditions for ambient vibration tests are simulated by generating time series of Gaussian white noise applied at each. The resulting absolute velocities at the nodes are considered as ambient vibration records under white noise conditions.

### 2.1. System Identification—Frequency Domain Approach

In this study, the frequency domain decomposition (FDD) method was chosen for the system identification analysis of ambient vibration measurements given its high computational efficiency and good accuracy (18). This method is summarized briefly below.

Considering a multiple degrees of freedom (MDOF) system with N nodes, the response $x_{j}\left(t\right)$ at node $j\in \left[1,\;2,\;\ldots ,N\right]$ in the time domain can be expressed in the frequency domain as
(1)$$X_{j}\left(\omega \right)=\int_{0}^{T}x_{j}\left(t\right)e^{-i2\pi \omega t}dt$$

The cross-spectral density (CSD) of the above response is given by
(2)$$S_{j,k}=E\left[X_{j}^{*}\left(\omega \right)X_{k}\left(\omega \right)\right]$$
where *E* is the expected value operator and * represents the complex conjugate.

The first step is to construct the cross-power spectral density (CPSD) matrix from the cross-power densities of all the response channels,
(3)$$S_{XX}\left(\omega \right)=E\left[\underline{X}\left(\omega \right){\underline{X}\left(\omega \right)}^{H}\right]$$
where
(4)$$\underline{X}\left(\omega \right)=\left\{\begin{array}{c} X_{1}\left(\omega \right)\\ \begin{array}{c} X_{2}\left(\omega \right)\\ \vdots \\ X_{N}\left(\omega \right) \end{array} \end{array}\right\}\;\mathrm{and}\;{\underline{X}\left(\omega \right)}^{H}=\left\{\begin{array}{cc} X_{1}^{*}\left(\omega \right) & \begin{array}{ccc} X_{2}^{*}\left(\omega \right) & \ldots & X_{N}^{*}\left(\omega \right) \end{array} \end{array}\right\}$$

The diagonal terms of matrix $S_{XX}\left(\omega \right)$ are the auto-spectral density (ASD) of each channel, and the off-diagonal terms are the CSD of responses. Next, the CPSD matrix representing the MDOF system is split into corresponding single degree of freedom (SDOF) systems by taking the singular value decomposition (SVD) of the system CPSD matrix at each discrete frequency $\omega_{i}$ [28]:(5)$$S_{XX}\left(\omega_{i}\right)=\left[U_{i}\right]\left[S_{i}\right]{\left[U_{i}\right]}^{H}$$
where $\left[U_{i}\right]=\left[\begin{array}{ccc} u_{1}^{i} & u_{2}^{i} & \begin{array}{cc} \cdots & u_{N}^{i} \end{array} \end{array}\right]$ is a matrix comprising singular vectors and $\left[S_{i}\right]$ is a diagonal matrix of singular values, which are often sorted from highest to lowest. A peak in the plots of singular values against frequency usually corresponds to a mode, and the first singular vector at that frequency is an estimate of the corresponding mode shape.
(6)$$\underline{\phi }={\underline{u}}_{1}^{i}$$

The mode shapes are complex valued with amplitude and phase angle. In case of lightly damped structures, the phase angles are close to 0 or 180 degrees.

### 2.2. Measurement Schemes

The first step in planning an output-only modal testing involves the selection of the number of nodes and/or degrees of freedom (DOFs) of the numerical model that will represent the test structure. This selection is largely governed by the number and availability of sensors and data acquisition systems (or data recording stations), the number and length of connecting cables and most importantly, the locations that can be accessed in the structure.

Another important assumption made in the analysis of output-only modal testing data is that the stiffness of the gravity load support system (beams and slabs) is much higher than that of the lateral load resisting system (columns and shear walls). As a consequence, the floors are often assumed to be rigid. The complete model is used for the placement scheme in tall buildings when the detection of torsional modes is important. The nodes selected for measurement are then often located at the extreme ends of the building.

### 2.3. Simplified Lumped Mass Line Models

The simplest placement scheme that minimizes the time and cost of output-only modal testing is the lumped mass staircase or line model, as explained in the first section of this article. It consists of a node to represent one or more floors whose mass is lumped at that node, and the frame members connecting each node represent the stiffness of all the vertical columns (or lateral load resisting system) connecting the floors. This model is the simplest approximation of a more detailed 3D model or, more precisely, the nodes of the line model usually represent the master nodes of each floor in a detailed 3D model.

In this study, the accuracy of the line model is compared to the complete 3D model for the mode shapes and mode frequencies. The most common sensors used in output-only modal testing are those that detect translations in horizontal and/or vertical directions, i.e., either displacement meters, velocity meters or, in some cases, accelerometers. Hence, it is not possible to measure the rotational response at any node. Combining this limitation with the geometrical limitations of the line model, it can be extremely challenging to estimate the properties of the torsional modes. In order to capture the torsional response of the floors, the sensors must be placed at locations where the torsional mode of the building produces large horizontal displacements. For buildings, any point at the periphery of the building, and more preferably in one of the corners, satisfies this condition.

The resulting measured model is a lumped mass stick model that coincides with a line passing vertically through the corner of the building, hence the name corner-line model. The corner-line model is expected to capture the translational as well as the torsional motions of the floors in the building.

### 2.4. Data Analysis of the Numerical Simulations

#### 2.4.1. Symmetrical Numerical Model

The output-only modal test comparisons are performed using the 3D measurement, the center-line (staircase) measurement, and the proposed corner-line measurement approach. The sensing nodes (or master nodes) are shown in red dots in each of the three cases as shown in Figure 1, while all the other nodes, referred to as slave nodes, are non-sensing nodes while their translations are extrapolated based on the translation of master nodes. Figure 2 compares the singular value (SV) plots of spectral densities of the 3D model, the center-line model and the corner-line model.

It can be readily noted that the SV plots of the corner-line model are in close agreement with the 3D model. Both SV plots show the torsional resonant frequencies represented by T1, T2, T3, T4 and T5. The center-line model, on the other hand, does not have torsional frequencies as evident from the respective SV plots shown in Figure 2, which only shows the translational modes given by the prefix ‘X’ or ‘Y’. This is because the translational motion due to torsion is null in a symmetric building at the center line.

Table 1 lists and compares the mode shapes for the three measurement schemes for the symmetric numerical model. The comparison indicates a good agreement between the mode shapes of all the measurement schemes when a mode is detected in all the three schemes (Figure 3). Note also that there is very little loss of accuracy in the estimates for higher modes as shown in Figure 3.

#### 2.4.2. Asymmetric Numerical Model

Next, the asymmetrical numerical model is used for the proposed three measurement schemes. The sensing nodes are identified by small red triangles as shown in Figure 4. The comparison of the singular values (SV) of the spectral densities for the 3D model, the center-line model and the corner-line model for the asymmetric numerical model is given in Figure 5. The translational mode resonant frequencies are identified with prefix X or Y, while the torsional mode resonant frequencies are shown with prefix T.

For the asymmetric model, the torsional frequencies are detected in both the center-line and corner-line models. This occurs because the center line does not coincide with the geometric center of the building, or more precisely, the center of rigidity of the building. However, the mode shapes associated with torsion are not clearly identified and tend to be very similar to the nearest translational mode. Table 2 lists and compares the mode shapes for the three measurement schemes for the asymmetric numerical model.

Figure 6a shows the modal assurance criterion matrix comparing modes of the asymmetric numerical model obtained by 3D modal model to those obtained by the corner-line model. The comparison clearly shows close agreement between the mode shapes of the 3D model and the corner-line model, which shows that the corner-line model can successfully be used to calculate mode shapes of a building, including torsional modes. The comparison of the mode shapes from the 3D modal model and that of the center-line is also showing good agreement, although it is not as close as the corner-line model.

## 3. Field Testing

In order to test the effectiveness of the corner-line model, ambient vibration tests were performed on two reinforced concrete shear wall buildings. The first building, designated RCSW1, constructed in 1965, is a medium-rise building located in downtown Montreal, Quebec, Canada. The floor plan of the building is fairly symmetric with dimensions of 26 m × 31 m approximately. Figure 7 shows a typical floor plan of the RCSW1 building. The building has six stories above ground level and one basement floor. Ambient vibration measurements were performed on three points on all six floors, but only one point was measured in the basement. Figure 7 also shows the area available for the test and location of three points chosen for measurement on all floors.

The second building, designated RCSW2, constructed in 1998, is a medium-rise building located in downtown Montreal, Quebec, Canada. The floor plan of the building is not symmetric and has dimensions of 25 m × 38 m approximately. Figure 8 shows a typical floor plan of the RCSW2 building. The building has five stories above ground level and one basement floor. Ambient vibration measurements were performed on three points on all six floors, but only one point was measured in the basement as shown. Figure 8 also shows the area available for testing and the location of three points chosen for measurement on all floors.

### 3.1. Ambient Vibration Tests

The AVTs on the two buildings were conducted using two sensors (Lennartz Velocity-meter) each with a separate City Shark data acquisition system (DAS). The two DAS were first synchronized using GPS and in the setups on the top floors, where reference and roving sensors were on the same floor; the start of the measurement was triggered by a radio-controlled starter (RCS) to ensure synchronous data recording. On all the other floors, the start of measurements could not be triggered simultaneously using the RCS because the radio waves could not travel through the floors; hence, data recording was started manually at preselected timestamps. At the end of the test, a dummy record was also taken with the two DAS next to each other while triggering the measurement with RCS. The timestamps in the data files for the first setup and the last dummy setup (both started with the RCS) were compared to determine the clock offsets and drifts between the two DAS for the duration of the test. The timestamps of the setups started manually were adjusted by proportionally distributing the drift according to the time of the measurements. The adjusted timestamps from the setups started manually were used to determine the synchronous portion of the data for the analysis. For a detailed discussion of synchronization issues, the readers are referred to Saeed, Chouinard and Sajid [29].

The data obtained from the AVTs were used to generate two measurement schemes for each of the two tested buildings. In the first scheme, all measured setups were included, which led to a frame model capable of defining the complete motion of the floors. In the second measurement scheme, only the reference node and those directly below it were considered. All other setups were ignored for the second measurement. This scheme resulted in a corner-line model, as described in above sections.

The two measurement schemes for both buildings were processed using the FDD to extract modal frequencies and mode shapes. The results of these analyses are presented below.

### 3.2. Modal Parameter Estimation

Figure 9 shows the comparison of the SV plots of spectral densities of the two measurement schemes used for the RCSW1 building. The SV plots for both measurement schemes are quite similar, and up to six frequencies are detected. Figure 10 shows the comparison of the SV plots of spectral densities of the two measurement schemes used for the RCSW2 building. The SV plots for both measurement schemes are quite similar, and only three frequencies are detected.

Table 3 tabulates the MAC matrix to compare the mode shapes of the 2D frame model and the corner-line model for the RCSW1 building. The MAC values close to 1.0 (highlighted in green) suggest that there is a very good correlation between the mode shape estimates from the two models. Table 4 shows the comparison of mode shapes of the 3D measurement scheme and corner-line measurement scheme. It can be observed that the mode shapes obtained by both measurement schemes are very similar. This suggests that the measurement time for the RCSW1 building could have been reduced to almost one third by measuring only one point on the corner of each floor without losing modal information.

Table 5 tabulates the MAC matrix to compare the mode shapes of the 2D frame model and the corner-line model for the RCSW2 building. Low values of MAC for two of the modes (highlighted in green) suggest that there is a poor correlation between the mode shape estimates from the two models. Table 6 shows the comparison of mode shapes of the 3D measurement scheme and corner-line measurement scheme. It can be observed that the mode shapes obtained by both measurement schemes are not similar except for the first mode where the mode shape comprises only translational motion.

These results suggest that the corner-line model can potentially give deficient modal information in buildings with an asymmetric plan (or asymmetric distribution of stiffness), because the mode shapes of such buildings have torsional components in the translational modes and vice versa, and one point may not be enough to accurately describe the motion of the whole floor.

Figure 11 shows the graphical representation of MAC values comparing the 2D frame model and corner-line models for RCSW1 and RCSW2 buildings. High values on the diagonal of MAC matrix of the RCSW1 building suggest that the corner-line model can be used to economize the placement scheme, while those of the RCSW2 building suggest that the corner-line model may not be suitable for measuring buildings with asymmetric plans (or an asymmetric distribution of stiffness).

## 4. Conclusions

The level of accuracy of modal models derived from output-only modal analysis using ambient vibration testing depends on the intended purpose of the model and on the accessibility of the nodes for measurement. When a large number of buildings have to be tested, for example for making seismic vulnerability maps of a city or a locality, then often, a compromise is made on the resolution of the modal model in order to minimize the time and funds required for completion of the tests on multiple buildings. The most commonly used modal model in such applications is the lumped mass stick model (or center-line model) with sensors placed at the center or as close to the center of the building as possible. This is helpful because the central part of the building is typically most easily accessible on all floors. In this research, it is shown that the modal model generated with the center-line sensing nodes may be lacking information on the torsional modes. However, if sensors in the line model are placed far away from the center, possibly at the extreme corners of the building, resulting in the so-called corner-line model, then the recorded output will have torsional modes adequately expressed in the data, and the torsional frequencies and mode shapes are easily detected in the analysis. Numerical simulations and testing on actual buildings are performed to demonstrate the validity of the proposed sensor placement approach. The results of the numerical simulations and field testing show that the modal parameter estimates are in close agreement when the output-only modal testing is performed using the frame model and the corner-lines sensing nodes for buildings which are symmetrical. For buildings with asymmetry in geometry or stiffness, the proposed sensing topology to obtain modal information similar to a frame model may not be achieved.

## Figures and Tables

**Figure 1 sensors-24-05687-f001:**
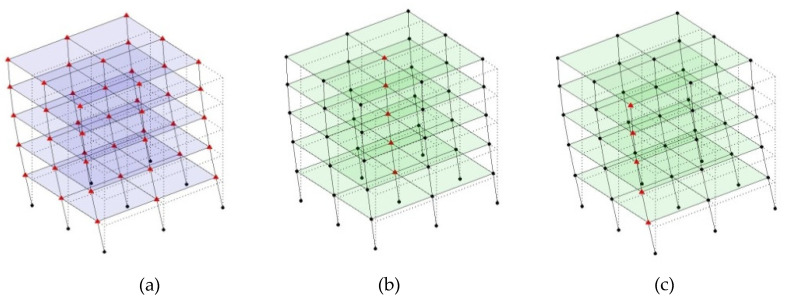
Symmetrical numerical model showing sensing nodes in small red triangles: (**a**) 3D, (**b**) center-line or staircase model, (**c**) proposed corner-line model.

**Figure 2 sensors-24-05687-f002:**
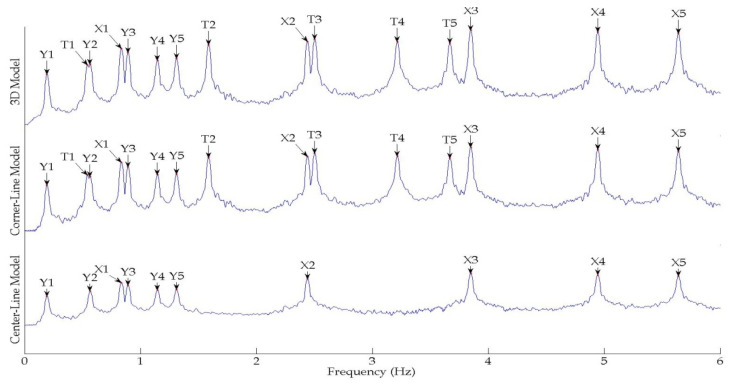
First singular value plots of spectral densities of various measurement schemes of the symmetrical numerical model.

**Figure 3 sensors-24-05687-f003:**
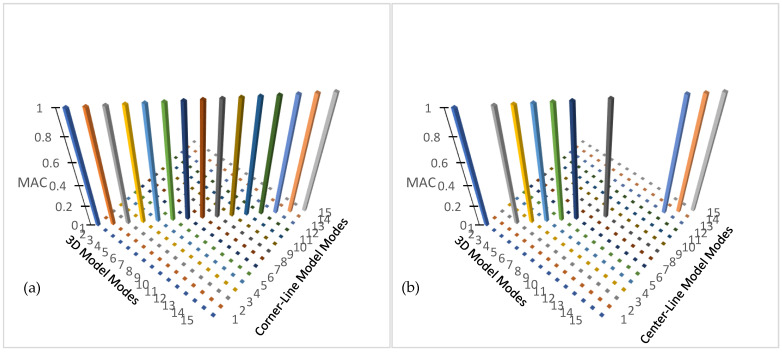
Modal assurance criterion showing comparison of mode shapes of symmetrical model for (**a**) 3D model and corner-line model, and (**b**) 3D model and center-line model.

**Figure 4 sensors-24-05687-f004:**
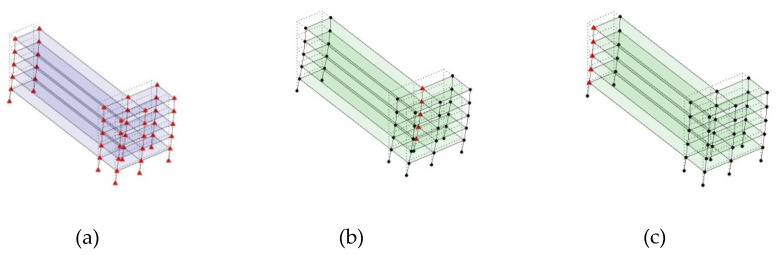
Sensing nodes for each measurement scheme: (**a**) 3D, (**b**) center-line model, (**c**) corner-line model.

**Figure 5 sensors-24-05687-f005:**
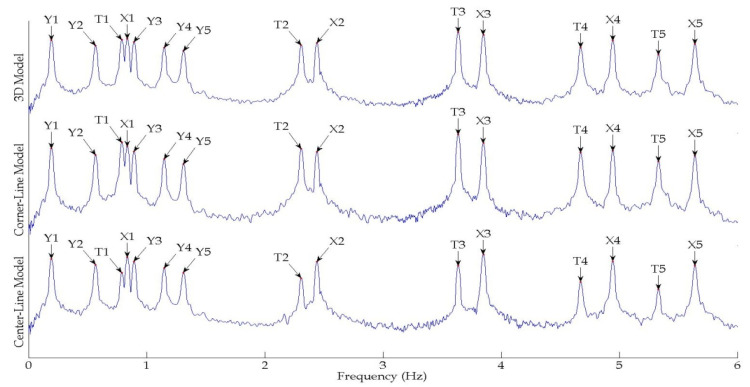
First singular value plots of spectral densities of various measurement schemes of the asymmetric numerical model.

**Figure 6 sensors-24-05687-f006:**
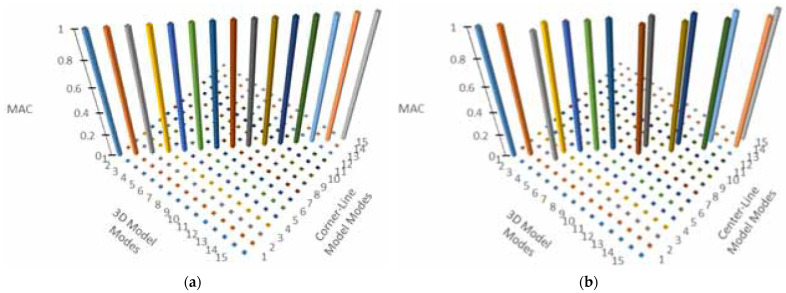
Model assurance criterion showing mode shape comparisons for asymmetrical numerical model between (**a**) 3D model and corner-line model, and (**b**) 3D model and center-line model.

**Figure 7 sensors-24-05687-f007:**
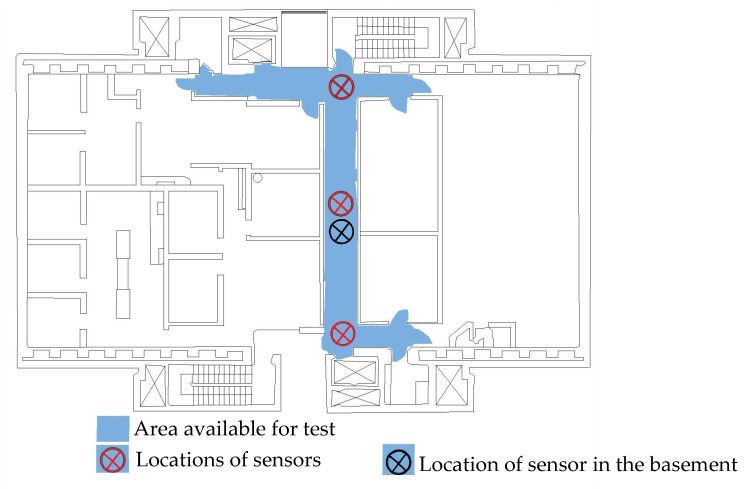
Typical floor plan of RCSW1, showing area available for testing and locations chosen for placement of sensors.

**Figure 8 sensors-24-05687-f008:**
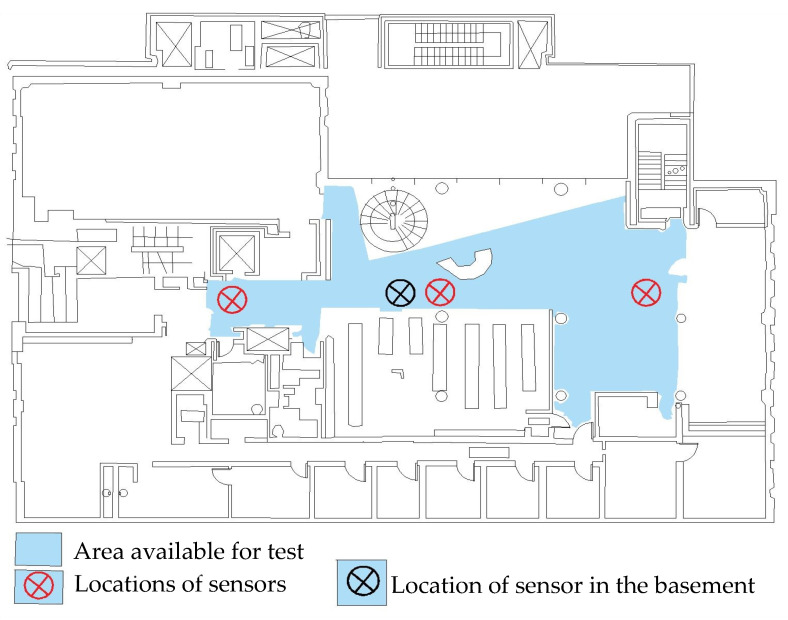
Typical floor plan of RCSW2, showing area available for testing and locations chosen for placement of sensors.

**Figure 9 sensors-24-05687-f009:**
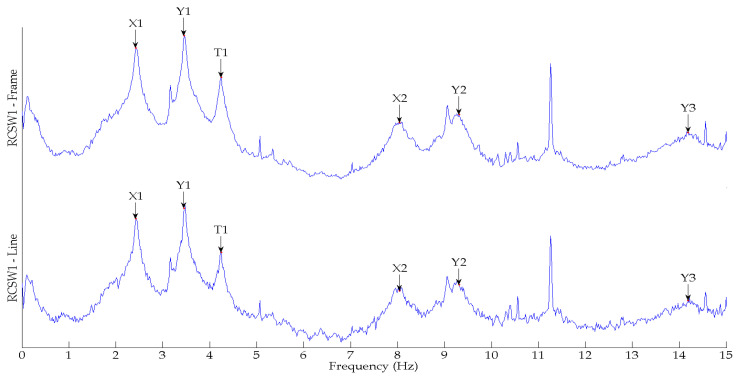
First singular value plots of spectral densities obtained from frame model and corner-line model of RCSW1.

**Figure 10 sensors-24-05687-f010:**
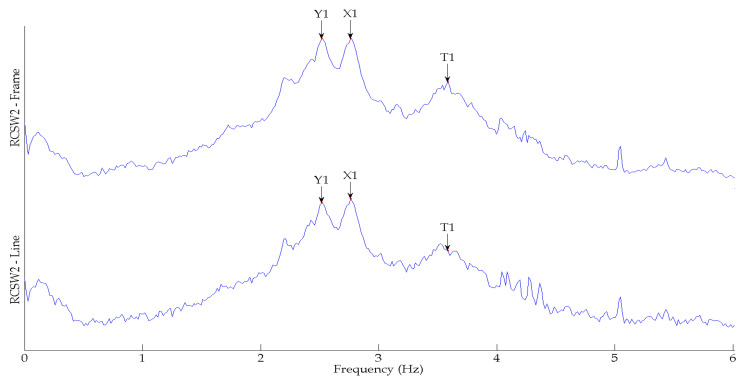
First singular value plots of spectral densities obtained from frame model and corner-line model of RCSW2.

**Figure 11 sensors-24-05687-f011:**
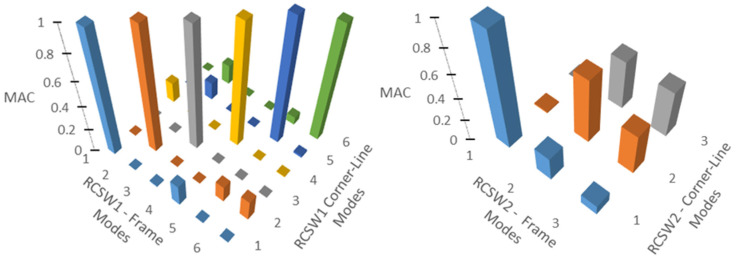
MAC matrix for comparison of mode shapes from frame models and corner-line models of RCSW1 and RCSW2 buildings.

**Table 1 sensors-24-05687-t001:** Comparison of mode shapes for different measurement schemes on the symmetrical numerical model.

Mode	Frequency	Mode Shape Comparisons		
		3D Model	Center-Line Model	Corner-Line Model
Y Translation 1	0.1938 Hz	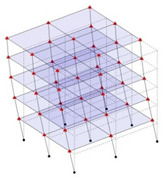	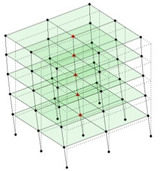	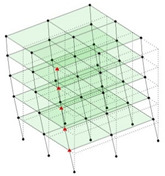
Torsion 1	0.5432 Hz	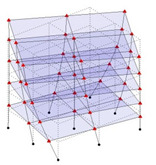	Not Detected	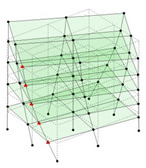
Y Translation 2	0.5661 Hz	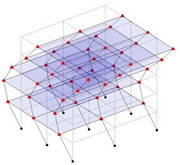	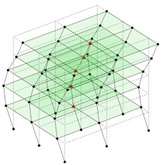	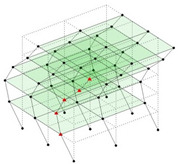
X Translation 1	0.8362 Hz	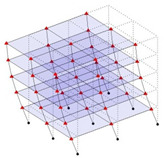	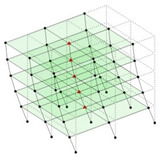	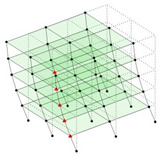
Y Translation 3	0.8942 Hz	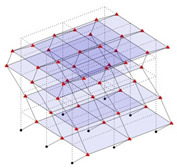	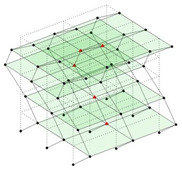	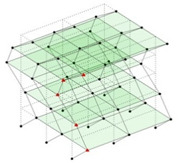
Y Translation 4	1.1475 Hz	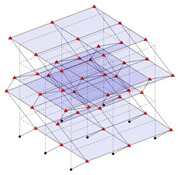	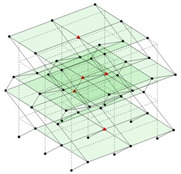	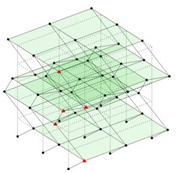
Y Translation 5	1.3123 Hz	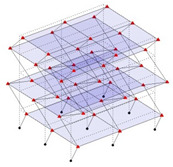	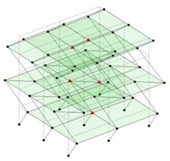	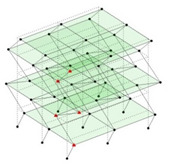
Torsion 2	1.5884 Hz	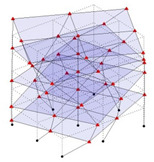	Not Detected	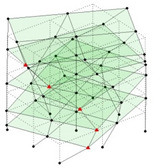
X Translation 2	2.4414 Hz	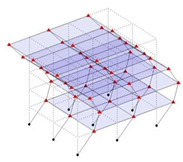	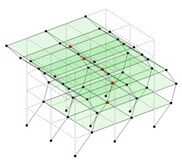	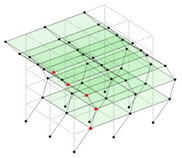
Torsion 3	2.5040 Hz	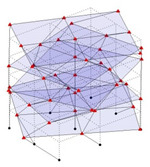	Not Detected	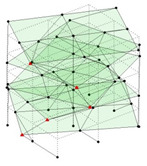
Torsion 4	3.2166 Hz	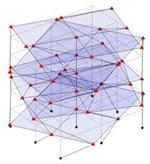	Not Detected	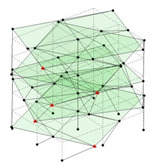
Torsion 5	3.6697 Hz	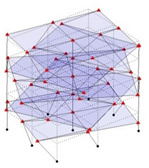	Not Detected	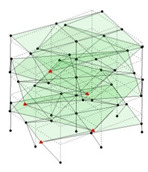
X Translation 3	3.8498 Hz	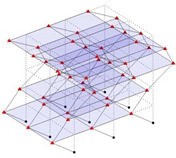	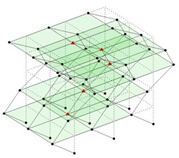	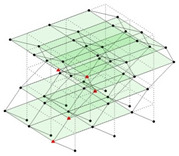
X Translation 4	4.9454 Hz	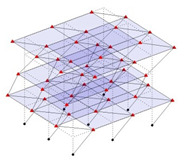	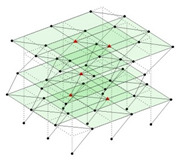	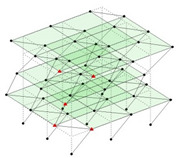
X Translation 5	5.6412 Hz	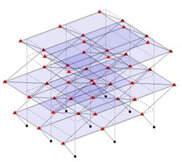	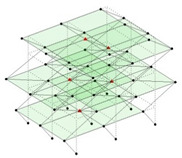	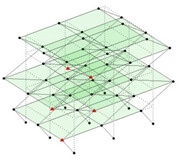

**Table 2 sensors-24-05687-t002:** Comparison of mode shapes obtained for the 3D model, center-line and the corner-line model for the first 15 modes.

Mode	Frequency	Mode Shape Comparisons		
		3D Model	Center-Line Model	Corner-Line Model
Y Translation 1	0.1938 Hz	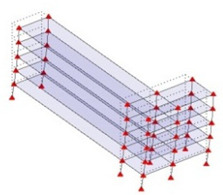	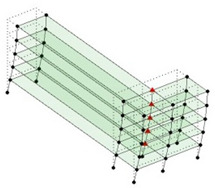	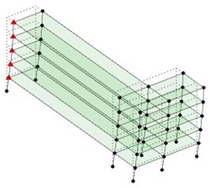
Torsion 1	0.7904 Hz	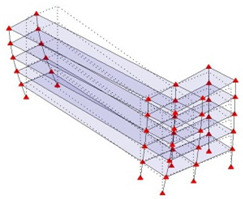	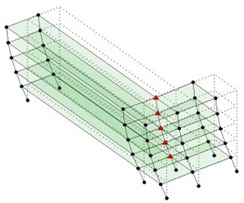	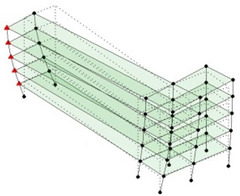
Y Translation 3	0.8942 Hz	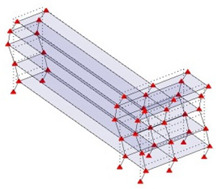	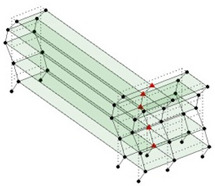	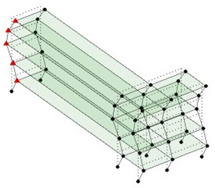
Y Translation 5	1.3123 Hz	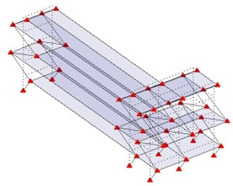	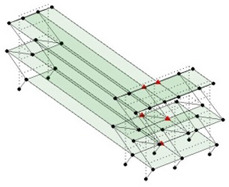	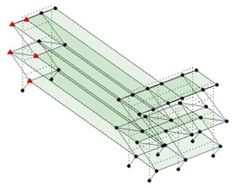
X Translation 2	2.4414 Hz	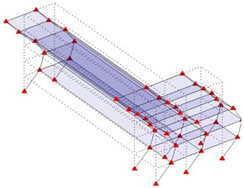	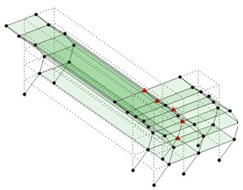	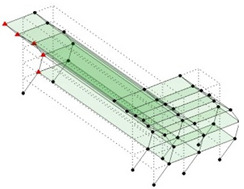
X Translation 3	3.8498 Hz	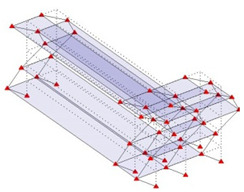	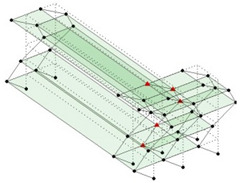	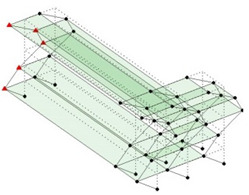
X Translation 4	4.9454 Hz	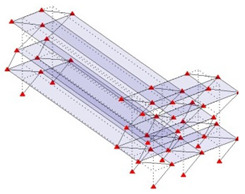	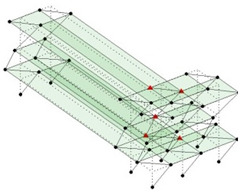	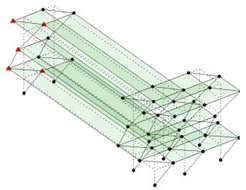
X Translation 5	5.6412 Hz	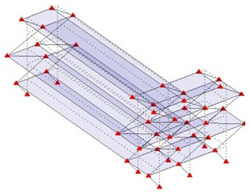	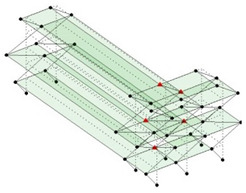	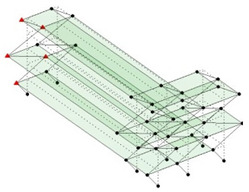
Y Translation 2	0.5676 Hz	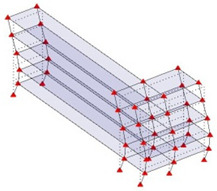	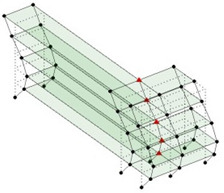	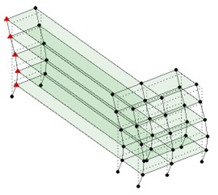
X Translation 1	0.8362 Hz	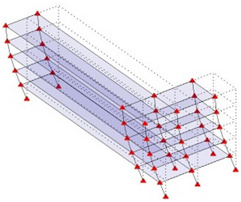	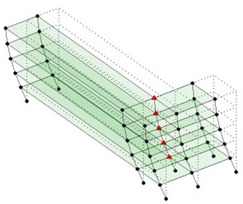	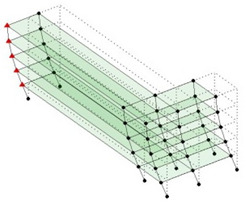
Y Translation 4	1.1490 Hz	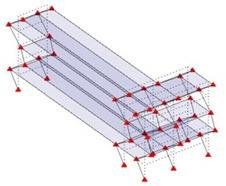	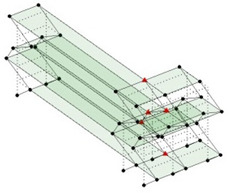	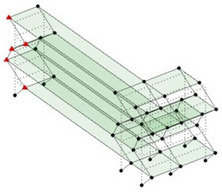
Torsion 2	2.3087 Hz	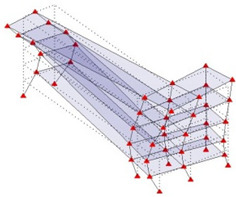	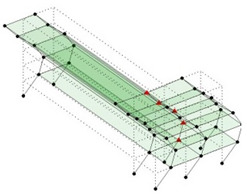	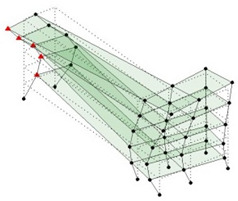
Torsion 3	3.6377 Hz	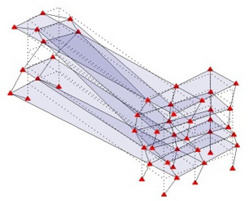	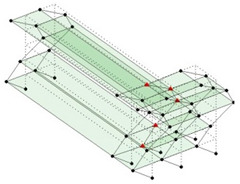	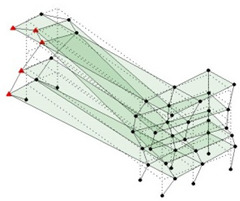
Torsion 4	4.6738 Hz	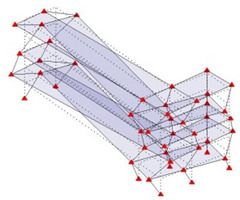	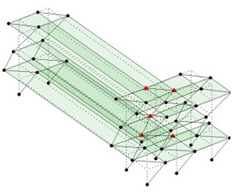	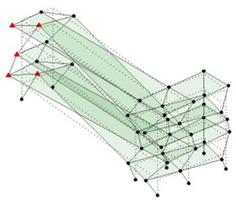
Torsion 5	5.3329 Hz	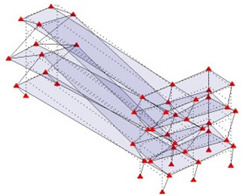	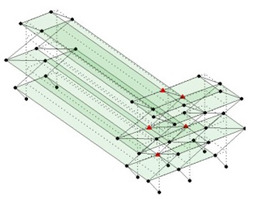	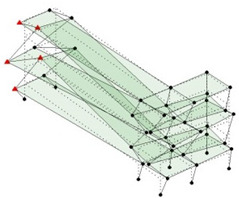
X Translation 5	5.6412 Hz	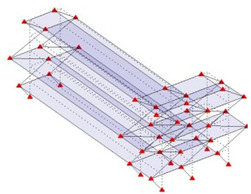	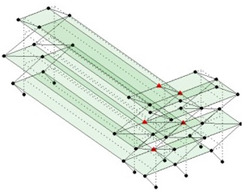	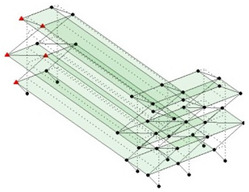

**Table 3 sensors-24-05687-t003:** MAC matrix comparing correlations of mode shapes from frame model and corner-line model of RCSW1.

		Modes of Corner-Line Model					
	Frequency (Hz)	2.426	3.448	4.242	8.041	9.308	14.191
Modes of Frame Model	2.426	0.992	0.000	0.010	0.179	0.000	0.004
	3.448	0.000	0.997	0.001	0.001	0.138	0.177
	4.242	0.006	0.002	0.977	0.000	0.000	0.000
	8.041	0.167	0.000	0.001	0.974	0.000	0.005
	9.308	0.000	0.144	0.000	0.003	0.989	0.017
	14.191	0.000	0.161	0.000	0.000	0.065	0.920

**Table 4 sensors-24-05687-t004:** Mode shapes of RCSW1 building obtained from frame model and corner-line model.

Mode	Frequency	Mode Shape Comparisons	
		Frame Model Measurement Scheme	Corner-Line Measurement Scheme
X Translation 1	2.426 Hz	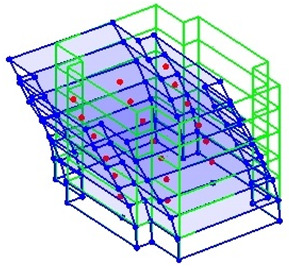	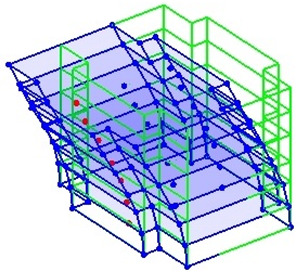
Y Translation 1	3.448 Hz	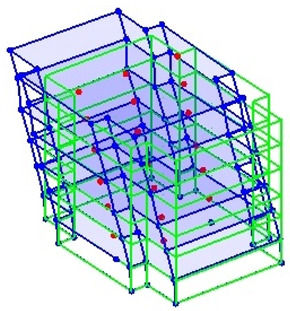	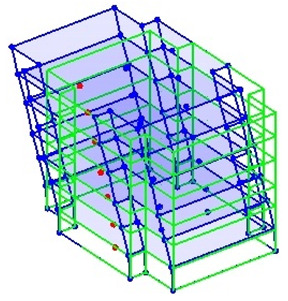
Torsion 1	4.242 Hz	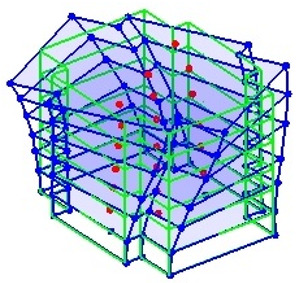	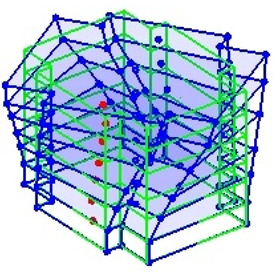
8.041 Hz	X Translation 2	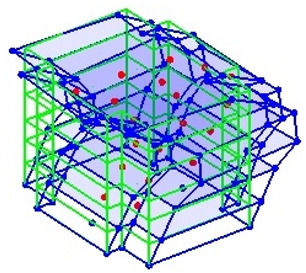	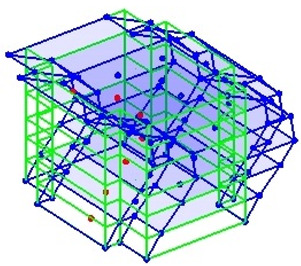
9.308 Hz	Y Translation 2	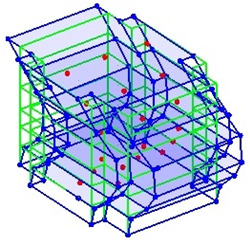	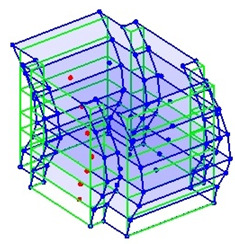
14.191 Hz	Y Translation 3	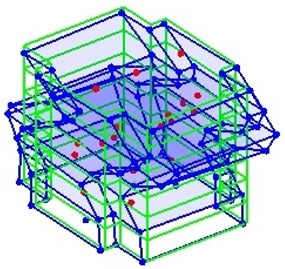	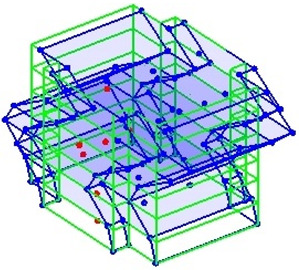

**Table 5 sensors-24-05687-t005:** MAC matrix comparing correlations of mode shapes from the frame model and corner-line model of RCSW2.

		Modes of Corner-Line Model		
	Frequency (Hz)	2.518	2.762	3.586
Modes of Frame Model	2.518	0.980	0.209	0.086
	2.762	0.012	0.568	0.362
	3.586	0.003	0.426	0.411

**Table 6 sensors-24-05687-t006:** Mode shapes of RCSW2 building obtained from the frame model and corner-line model.

Mode	Frequency	Mode Shape Comparisons	
		Frame Model Measurement Scheme	Corner-Line Measurement Scheme
Y Translation 1	2.518 Hz	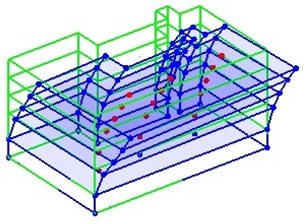	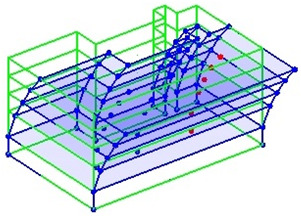
X Translation 1	2.762 Hz	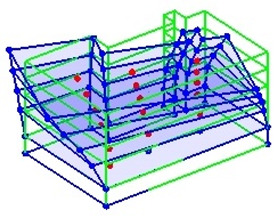	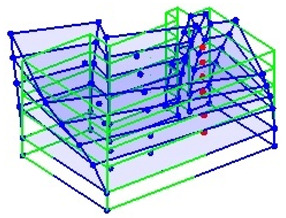
Torsion 1	3.586 Hz	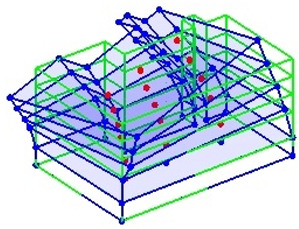	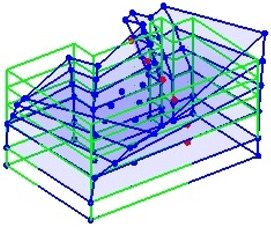

## Data Availability

The data can be made available upon reasonable request and subject to permission from the owners of the structures at which the measurements were performed.

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
