# Peer review of "Optimal Sensor Placement for Enhanced Efficiency in Structural Health Monitoring of Medium-Rise Buildings"

_sensors, 2024, doi:10.3390/s24175687_

Round 1

Reviewer 1 Report

Comments and Suggestions for Authors

This study proposed a solution of deploying sensors using diagonal lines. For plane-symmetrical buildings, the number of sensors used was reduced while ensuring the quality of modal extraction, which greatly reduced the cost and had great engineering application value. The manuscript would be acceptable for publication if the authors address the following questions. All answers should be included in the manuscript.

1.     In Figure 9, what do the peaks between X2 and Y2, and between Y2 and Y3 represent? Can you explain?

2.     In complex structures, multimodal superposition is a common occurrence. How does the diagonal arrangement of sensors in this article decompose the structural modes under natural environmental excitation?

3.     Is the arrangement proposed in this article limited to simpler structures to achieve better performance? Please explain clearly.

4.     Figure 3 quality is suboptimal, and the image resolution needs to be increased to see the text clearly.

5.     The layout of Table 1, Table 2, and Table 4 needs to be adjusted.

6.     There are problems with the section numbering, which needs to be unified throughout the text.

7.     The number of references within five years is small, please supplement them.

Comments on the Quality of English Language

The language and style of the manuscript are generally clear and concise, effectively communicating the key ideas and findings.

Reviewer 2 Report

Comments and Suggestions for Authors

This study explores the optimal placement of sensors to optimize their efficiency in Structural Health Monitoring of Medium-rise Buildings by Ambient Vibration Tests. The manuscript presents parametric studies of two 5-store buildings. Further, Ambient Vibration Tests were conducted in two buildings: one with a relatively symmetrical floor plan and another with an asymmetrical one. The research was well-designed and executed, and the methodology was well-written but needed some clarification before publishing.

a) The manuscript (line 29 o) contains typos and formatting issues. Also, there are systematically double spaces within the manuscript. Please check and revise carefully. 

b) The abbreviation should be mentioned once, the first time it will be used in the manuscript.

c) The introductory section should provide a comprehensive overview of the importance of Structural Health Monitoring (SHM) in the context of reinforced concrete structures. Furthermore, it should include a paragraph dedicated to the application of diverse Structural Health Monitoring techniques, such as the Electro-Mechanical Impedance Method (10.3390/s22249592, 10.3390/s24020386), before an in-depth discussion of the ambient vibration technique.

d) In lines 238 - 240, the authors state, "The building has 5 stories above ground level and one basement floor. Ambient Vibration measurements were performed on three points on all six floors, but only one point was measured in the basement, as shown."  However, the sensor's location in the basement is not marked in Figure 8. Please revise Figure 7 and Figure 8 to incorporate the sensor's location in the basement of the building RCSW1 and RCSW2, respectively.

Comments on the Quality of English Language

Despite being well-written in English and carefully edited, the manuscript contains a few minor typos. Please check the manuscript carefully.

Round 2

Reviewer 2 Report

Comments and Suggestions for Authors

The authors satisfactorily revised the manuscript, which can be published in the present form.